# Effect of Surgical Release of Entrapped Peripheral Nerves in Sensorimotor Diabetic Neuropathy on Pain and Sensory Dysfunction—Study Protocol of a Prospective, Controlled Clinical Trial

**DOI:** 10.3390/jpm13020348

**Published:** 2023-02-17

**Authors:** Simeon C. Daeschler, Anna Pennekamp, Dimitrios Tsilingiris, Catalina Bursacovschi, Martin Aman, Amr Eisa, Arne Boecker, Felix Klimitz, Annette Stolle, Stefan Kopf, Daniel Schwarz, Martin Bendszus, Ulrich Kneser, Zoltan Kender, Julia Szendroedi, Leila Harhaus

**Affiliations:** 1Department of Hand, Plastic and Reconstructive Surgery, Burn Center, BG-Trauma Center Ludwigshafen/Rhine, Department of Hand and Plastic Surgery, University of Heidelberg, 67071 Ludwigshafen, Germany; 2Department of Internal Medicine 1 and Clinical Chemistry, University Hospital of Heidelberg, 69120 Heidelberg, Germany; 3German Center for Diabetes Research, 85764 Neuherberg, Germany; 4Department of Neuroradiology, Heidelberg University Hospital, 69120 Heidelberg, Germany; 5Joint Heidelberg-ICD Translational Diabetes Program, Helmholtz-Zentrum, 85764 Neuherberg, Germany; 6Department of Handsurgery, Peripheral Nerve Surgery and Rehabilitation, BG Trauma Hospital, 67071 Ludwigshafen, Germany; 7Department of Orthopedic Surgery, Section Upper Extremity, University Hospital Heidelberg, 69120 Heidelberg, Germany

**Keywords:** diabetes, diabetic (poly-)neuropathy, nerve compression, compression neuropathy, nerve entrapment, pressure measurement, pressure sensor

## Abstract

Background: Nerve entrapment has been hypothesized to contribute to the multicausal etiology of axonopathy in sensorimotor diabetic neuropathy. Targeted surgical decompression reduces external strain on the affected nerve and, therefore, may alleviate symptoms, including pain and sensory dysfunction. However, its therapeutic value in this cohort remains unclear. Aim: Quantifying the treatment effect of targeted lower extremity nerve decompression in patients with preexisting painful sensorimotor diabetic neuropathy and nerve entrapment on pain intensity, sensory function, motor function, and neural signal conduction. Study design: This prospective, controlled trial studies 40 patients suffering from bilateral therapy-refractory, painful (*n* = 20, visual analogue scale, VAS ≥ 5) or painless (*n* = 20, VAS = 0) sensorimotor diabetic neuropathy with clinical and/or radiologic signs of focal lower extremity nerve compression who underwent unilateral surgical nerve decompression of the common peroneal and the tibial nerve. Tissue biopsies will be analyzed to explore perineural tissue remodeling in correlation with intraoperatively measured nerve compression pressure. Effect size on symptoms including pain intensity, light touch threshold, static and moving two-point discrimination, target muscle force, and nerve conduction velocity will be quantified 3, 6, and 12 months postoperatively, and compared (1) to the preoperative values and (2) to the contralateral lower extremity that continues non-operative management. Clinical significance: Targeted surgical release may alleviate mechanical strain on entrapped lower extremity nerves and thereby potentially improve pain and sensory dysfunction in a subset of patients suffering from diabetic neuropathy. This trial aims to shed light on these patients that potentially benefit from screening for lower extremity nerve entrapment, as typical symptoms of entrapment might be erroneously attributed to neuropathy only, thereby preventing adequate treatment.

## 1. Introduction

Nowadays, 1 in 11 adults, or about 463 million people worldwide, are estimated to suffer from diabetes, a number that has more than tripled over the past 2 decades [1]. Consequently, approximately one-tenth of global health expenditures, equaling USD 760 billion, are spent on diabetes [1]. Up to 50% of those diabetic patients worldwide eventually develop peripheral diabetic neuropathy [2], rendering diabetes the most common cause of neuropathy to date [3].

The progressive neurodegeneration in diabetic neuropathy occurs in a length-dependent fashion and primarily affects sensory and autonomic axons of the peripheral nervous system [4]. This causes sensory dysfunction and, in one-third of the affected patients, severe and persistent pain [5,6,7]. Particularly, distal axon terminals including intraepidermal nerve fibers are affected first, proceeding to proximal axonal degeneration [5]. Although the pathomechanism of this characteristic pattern of neurodegeneration is subject to an ongoing debate, glucotoxicity, inflammation, progressive intraneural microangiopathy, and impaired neuronal and Schwann cell metabolism are likely involved [8,9,10,11,12].

Beyond these mechanisms, previous work indicated that progressive intra- and extraneural fibrotic tissue remodeling [13,14,15,16,17,18], similar to diabetes-associated fibrosis observed in multiple parenchymal organ systems, affect the mechanical properties of the nerve and surrounding tissues [19,20,21,22]. Accordingly, neurosonographic studies observed a significant increase in circumference and stiffening of peripheral nerves in patients with diabetic neuropathy, associated with an increase in connective tissue in the endoneurium [13,14,15,16,17,18]. This supports the idea that axonal degeneration secondary to diabetes may be promoted by mechanical nerve entrapment in some patients. Given the hyper-excitability and altered stimulus–response function in sensory neurons of diabetic individuals [23,24], external pressure on the neuropathic nerve may promote sensory dysfunction and pain in these patients.

In advanced diabetes, intraneural capillary dysfunction and consequently impaired oxygen extraction, combined with phenotypic alterations in mitochondrial biology in diabetic neurons, result in reduced adaptability to fluctuating energy demand and, thus, may cause exhaustion of the ATP supply in distal axonal components [9]. Chronic compression further reduces nerve perfusion and thereby its nutrient supply, promoting a hypoxic proinflammatory environment [10,11,25,26], thus, likely contributing to Schwann cell impairment and axonal injury at the compression site.

Therapeutic options are limited for this debilitating disease. In the United States, all clinical trials that aimed to prevent disease progression of diabetic neuropathy have failed so far [27]. Modifying symptom severity has been equally challenging, as even state-of-the-art pain medication may achieve an average individual pain reduction of only 30–50% in these patients [28,29]. The consequent enormous socioeconomic burden of this disease, at a personal and societal level, mandates interdisciplinary management of these patients, taking advantage of all available therapeutic options. This trial aims to investigate the surgical release of entrapped lower extremity nerves in selected patients with diabetic neuropathy as an adjunct therapeutic option to address the compressive component of this multifactorial disease and, therefore, may alleviate pain and improve sensory function.

## 2. Study Objectives

The primary objective of this trial is quantifying the treatment effect size of unilateral surgical release of the tibial nerve in the tarsal tunnel and/or the common peroneal nerve below the muscle fasciae in the proximal lower leg [30] on pain intensity, analgesics consumption, and sensory dysfunction in the skin area supplied by the decompressed nerves.

Secondary objectives are quantifying the treatment effect on target muscle force and fatigability, as well as the incidence of diabetic ulcers in the skin area supplied by the decompressed nerve. Furthermore, tissue biopsies from the perineural fat and overlying compressive fascia are analyzed to explore perineural tissue remodeling in correlation with intraoperatively measured tissue pressures in nerve constriction sites.

Trial registration: German Clinical Trials Register (DRKS); DRKS-ID: DRKS00021151.

## 3. Materials and Methods

### 3.1. Participants, Eligibility Criteria

A total of 40 type 1 and type 2 diabetic patients (age 18–90 years) suffering from diabetic sensorimotor neuropathy and showing clinical signs of compression of the tibial nerve in the tarsal tunnel and/or the common peroneal nerve below the muscle fasciae in the proximal lower leg on both lower extremities are included in this trial. Of those, *n* = 20 participants suffer from therapy-refractory pain with intensities of 5 and greater on the visual analogue scale (VAS), and another *n* = 20 participants suffer from therapy-refractory hypoesthesia, predominantly localized in the skin areas supplied by the potentially entrapped peripheral nerves.

Detailed eligibility criteria for participating in this trial are as follows:Age 18–90 years;Type 2 or type 1 diabetes mellitus diagnosed according to the guidelines of the German Diabetes Association [31].

And
Diabetic sensorimotor polyneuropathy of both lower extremities [32]:
-Neuropathy Deficit Score > 5 points;-Neuropathy Deficit Score 3–5 points + Neuropathy Symptom Score > 4 points.

And
Clinical evidence of bilateral local nerve compression of the tibial nerve in the tarsal tunnel and/or the common peroneal nerve under the muscle fascia of the peroneus longus muscle:
-Predominant focal (pain) symptoms in the innervation area of the nerve distal to the compression site;-Hoffman–Tinel’s sign over the compression site;-Characteristic electrodiagnostic findings of compression e.g., local conduction block;-Characteristic findings of compression in the MR-neurography, e.g., local T2 weighted hyperintensity of the nerve within the compression site.

And
Bilateral, therapy-refractory pain in the area supplied by the potentially entrapped nerve under conventional, guideline-compliant analgesic therapy (VAS >5);Bilateral hypesthesia in the area supplied by the potentially entrapped nerve defined as a tactile perception threshold greater than 6 g but less than 60 g.Exclusion criteria are as follows:HbA1c in the blood plasma > 8.5% (i.e., poor disease control);Reduced ability to give consent and/or legal capacity (e.g., due to mental illness);Contraindications to lower extremity surgery;Contraindications for electrodiagnostics, such as implanted cardiac pacemaker/defibrillator;Diseases that potentially confound the outcomes of interest, in particular:
-Other diseases of the peripheral or central nervous system not mentioned under “Inclusion criteria”;-Moderate or severe peripheral arterial disease (ankle–brachial index <0.75, or missing foot pulses from the dorsal pedis artery or posterior tibial artery on at least one lower extremity);-Dermatological conditions of the lower extremities.

### 3.2. Intervention

Patients with diabetic sensorimotor neuropathy of both lower extremities and signs of bilateral tibial nerve and/or peroneal nerve entrapment undergo unilateral decompression of the entrapped nerve(s) (Figure 1A–H) by an experienced senior microsurgeon (LHW). Intraoperatively, prior to surgical release, the nerve compression pressure is measured by inserting a CE-certified flexible pressure probe (Mikro-Cath, Millar, Houston, TX, USA) into the anatomical constriction site (Figure 1B,F,G).

A tourniquet is used to improve intraoperative visibility. Decompression of the tibial nerve in the tarsal tunnel is conducted through an 8–10 cm long skin incision on the inner malleolus (Figure 1A). The tarsal ligament is split and partially resected to avoid recurrence. The medial and lateral plantar nerves are then released as they pass under the deep fascia of the abductor hallucis muscle. The tibial nerve is carefully mobilized from the surrounding tissue and the decompression is verified by subsequent elevation of the nerve (Figure 1C) [30,33].

The common peroneal nerve is decompressed through a 5–7 cm long skin incision around the fibular head and neck (Figure 1D,E). The nerve is traced distally, and the compressing fasciae are split, namely the fascia of the peroneus longus muscle, the posterior and anterior crural intermuscular septum, the intermuscular septum between the extensor digitorum longus and tibialis anterior, and the fascia of the soleus muscle, including any accessory bands of connective tissue along the nerve (Figure 1H). Decompression is verified by subsequent elevation of the nerve.

Intraoperatively, tissue biopsies are harvested from (1) the tarsal ligament, (2) the fascia of the peroneus longus muscle, and (3) the perineurial tissue of the decompressed nerves(s) for subsequent analysis. A meticulous bipolar hemostasis is carried out with subsequent multi-layered skin closure using surgical sutures.

The surgery is performed as an inpatient procedure, using postoperative compressive dressings, daily wound care, and a slightly elevated position of the operated limb for 48 h post-surgery to reduce postoperative swelling and risk of hematoma.

### 3.3. Outcomes

#### 3.3.1. Pain Intensity

Participants are asked to quantify pain intensity using the numeric rating scale (NRS) in the skin areas supplied by the entrapped peripheral nerves in each lower extremity, preoperatively as well as 3, 6, and 12 months postoperatively.

#### 3.3.2. Sensory Function

Participants undergo sensory testing in the skin areas supplied by the tibial and peroneal nerves (Figure 2) in both lower extremities preoperatively, as well as 3 and 12 months postoperatively.

##### Tactile Sensory Threshold

The tactile sensory threshold describes the lowest detectable light touch stimulus. For this trial, it is assessed according to the protocol of the German Research Association for Neuropathic Pain (DFNS) [34] using a set of standardized CE-certified von Frey hairs (8, 16, 32, 64, 128, 256, and 512 mN). The geometric mean value of the tactile detection threshold is determined in five series using a modified limit value method (“method of limits”) based on decreasing stimulus intensities.

##### Two-Point Discrimination

Two-point discrimination describes the ability to recognize two stimuli as two individual stimuli (and not one stimulus) when they are simultaneously applied in close proximity. Static and dynamic two-point discrimination are assessed in a participant-blinded fashion using a two-point esthesiometer (NCD Medical/Prestige, Los Angeles, CA, USA) according to the protocol of Dellon and Mackinnon [35]. The esthesiometer is placed on the skin with its own weight (10 g), so that the skin is minimally deformed to a standardized extent. The dynamic two-point discrimination assessment follows the same protocol; however, the stimuli are slowly and uniformly moved over the skin in the examination area over a 1.5 cm distance, without applying additional pressure. Two-point discrimination is defined as the minimum distance between two stimuli in millimeter that is correctly recognized as two stimuli in two out of three applications.

##### Quantitative Sensory Testing (QST)

Additionally, participants undergo standardized quantitative sensory testing (QST) on both lower extremities according to the protocol of the German Research Network on Neuropathic Pain (DFNS) [36] preoperatively, and at least one year postoperatively. Complete QST determines neuropathic deficits in 13 different categories, including cold detection threshold, warm detection threshold, thermal sensory limen, cold pain threshold, heat pain threshold, pressure pain threshold, mechanical pain threshold, mechanical pain sensitivity, wind-up ratio, mechanical detection threshold, vibration detection threshold, dynamic mechanical allodynia, and paradoxical heat sensation.

#### 3.3.3. Motor Function

Maximum isometric muscle strength and fatigability of movements that involve the target muscles of the entrapped tibial and peroneal nerves are assessed in both lower extremities preoperatively, as well as 3 and 12 months postoperatively.

To assess for a potential therapeutic effect of surgical decompression of the tibial nerve in the tarsal tunnel, the maximum isometric muscle strength of great toe flexion is measured using a handheld dynamometer (MicroFET 2, Hoggan Scientific, Salt Lake City, UT, USA). The peroneal nerve’s target muscles are assessed using the maximum isometric muscle strength of great toe extension.

The dynamometer is placed perpendicular to the direction of the force vector and the application points of the dynamometer are standardized using anatomical landmarks. The patient is then instructed to perform the movement to be tested and to maximally press against the force absorbing plate of the dynamometer for a period of five seconds while avoiding explosive movements. The examiner stabilizes the limb proximal to the limb segment to be tested. Three repeat tests are then carried out with approximately 30 s rest in-between. The maximum isometric force mean and the peak values are documented [37].

Muscle fatigability and coordination are assessed using the one-legged toe stand test. Participants perform a repeated concentric–eccentric muscle contraction of the plantar flexors in a one-legged stand [38,39]. The cadence of the toe stands is acoustically specified with a metronome at 60 per minute. Subjects are instructed to maximally raise the heel on each repetition while keeping the knee extended and the trunk upright. The test ends when the subjects can no longer (completely) lift the standing heel or when they can no longer maintain the set pace, knee angle, or trunk position. The number of correct heel raises is counted for each leg and recorded.

#### 3.3.4. Nerve Conduction Velocity

Nerve conduction velocity of the tibial and peroneal nerves over the respective nerve compression site are quantified preoperatively and 6 to 12 months postoperatively for both lower extremities using adhesive surface electrodes and a CE-certified neurophysiology device (Viasys Healthcare VikingQuest^®^, Viasys Healthcare GmbH, Höchberg, Germany).

#### 3.3.5. Tissue Biopsies

Intraoperatively harvested tissue biopsies from the tarsal ligament and the fascia of the peroneus longus muscle, are formalin-fixed, stained, and analyzed for histomorphology, cell type distribution, and extracellular matrix composition to assess tissue remodeling. Perineural fat biopsies harvested from entrapment sites are snap frozen and archived into the Collaborative Research Centre (CRC) 1118 diabetes-specific biobank, part of the tissue bank of the National Center for Tumor Diseases (NCT). Collectively, these CRC 1118 biosamples will be analyzed for glucose metabolism pathologies, metabolic pathway regulation, characterization of cellular senescence, inflammation, and fibrosis to decipher the chronic tissue remodeling processes that potentially underlie nerve compression in this cohort.

### 3.4. Participant Timeline

Following initial recruitment, the inclusion and exclusion criteria are applied, and written consent is acquired from each participant. Then, participants undergo a standardized clinical exam and history taking to assess pain intensity, sensory dysfunction, motor function, and to detect neuropathic ulcerations in the skin innervation area of the tibial and/or peroneal nerve in both legs with subsequent electrodiagnostic testing. Nerve decompression surgery is performed, and in-person follow-up exams are scheduled 3 and 12 months post-surgery to repeat the standardized preoperative assessments (Figure 3). Six months post-surgery, a phone interview is conducted to assess pain intensity. Trial participation ends 12 months post-surgery.

### 3.5. Sample Size

Sample size calculation is based on effect sizes achieved in previous clinical studies [40,41,42]. By allocating both lower extremities of one participant as one observation unit, a sample size of *n* = 20 is required for painful and painless participants, respectively, to achieve a power of > 0.8 at a significance level of 5% with an expected drop-out rate of 20%.

### 3.6. Recruitment

Patients potentially suitable for this trial are primarily recruited through diabetic outpatient clinics of the University of Heidelberg, as well as through the peripheral nerve clinic at the BG trauma center, Ludwigshafen, University of Heidelberg.

### 3.7. Allocation and Blinding

The selection of the extremity to be operated on is made after inclusion in the study and completion of the pre-operative tests based on the patient’s preference. A randomized selection of the extremity that undergoes surgery was considered unethical given the often-asymmetrical pain distribution among both legs in our patient population.

Outcomes are assessed through a blinded examiner. The wounds/scars in the operated limb and the corresponding area of the contralateral extremity will be covered by a third person immediately before examination.

### 3.8. Data Management and Statistics

The collected data will be pseudonymized using case report forms with a unique participant number. Before publication and during statistical analysis the data will be completely anonymized. All collected data are analyzed using descriptive statistics (absolute and relative frequencies or mean, standard deviation, median, interquartile range (IQR), and, if necessary, confidence intervals), and graphic methods to characterize distributions. Because intervention and control group measurements are obtained from one subject and are collected in serial measurements over time, the data are analyzed as matched samples using signed-rank tests to address the following hypotheses.

The targeted surgical release of a mechanically compressed nerve in diabetic, sensorimotor polyneuropathy results in the following:A reduced pain intensity in the innervation area of the nerve distal to the decompression site 3, 6, and 12 months postoperatively, compared to the preoperative pain intensity and compared to an intraindividual non-decompressed control (contralateral lower extremity).A reduced analgesics consumption 3, 6, and 12 months postoperatively, compared to the preoperative baseline.Improved sensation (two-point discrimination, tactile detection threshold) in the innervation area of the nerve distal to the decompression site 3 and 12 months postoperatively compared to the preoperative situation and compared to an intra-individual, non-decompressed control (contralateral lower extremity).Increased isometric muscle strength in target muscles supplied by muscle branches arising from the decompressed nerve distal to the decompression site, 3 and 12 months postoperatively, compared to the preoperative situation and compared to an intra-individual, non-decompressed control (contralateral lower extremity)An increase in nerve conduction velocity across the decompression site 3 and 12 months postoperatively compared to the preoperative situation and compared to an intra-subject non-decompressed control (contralateral lower limb).A lower prevalence of neuropathic ulcerations in the innervated skin area of the nerve distal to the decompression site compared to an intra-individual, non-decompressed control (contralateral lower extremity) in the 12-month postoperative follow-up period.

## 4. Discussion

This trial aims to investigate a potential treatment effect of surgical nerve release in patients with diabetic neuropathy and lower extremity nerve entrapment on pain and sensory function. Secondary outcomes of interests are nerve conduction velocity, motor function, and prevalence of diabetic ulcerations.

Another objective of this study is to generate hypotheses on currently unknown aspects of the pathophysiology of nerve compression in diabetic sensorimotor polyneuropathy. Here, we focus on the chronically progressive tissue remodeling processes at the compression site by analyzing intraoperatively harvested tissue biopsies, and introduce, for the first time, tissue pressure measurements at the site of compression. The hypotheses generated from this trial will be addressed in subsequent studies in a confirmatory manner.

The currently available clinical studies on therapeutic nerve decompression in this cohort are limited by two essential aspects. First, a heterogeneous study collective with multiple compression and varying decompression sites of different nerves in the lower extremities were included [40,43,44,45]. Second, the evaluation of treatment success through postoperative pain reduction usually referred to the entire lower extremity, instead of a survey focused on the sensory innervation areas of the decompressed nerves. Therefore, a mechanistic causality of nerve decompression and the indicated pain reduction cannot be conclusively established. A pronounced placebo effect and/or altered extremity perfusion through the simultaneous release of multiple anatomical constriction sites may also interfere with study outcomes [46,47,48].

This trial aims for increased internal and external validity through a homogeneous patient collective and by using a standardized intervention with a standardized postoperative assessment of predefined target criteria. These data, together with intraoperative compression pressure measurements and morphological analysis of perineural tissue remodeling, may help to give insight into the potential role of nerve entrapment in patients with symptomatic diabetic neuropathy. However, it must be emphasized that surgical release may not address any underlying metabolic–toxic aspects of this disease but solely alleviates mechanical strain on entrapped neuropathic nerves. Due to the invasiveness of this procedure, and the risk of poor wound healing in diabetics, eligible patients need to be selected with great care. Partaking in the interdisciplinary management of patients with diabetic neuropathy, experienced nerve surgeons may help to allocate this treatment perspective responsibly to those likely to benefit from this intervention.

This trial aims to provide evidence on whether selected patients suffering from diabetic neuropathy with unusual distribution of pain or sensory dysfunction may benefit from referral to experienced nerve surgeons to reliably identify and address any potential nerve entrapment.

## Figures and Tables

**Figure 1 jpm-13-00348-f001:**
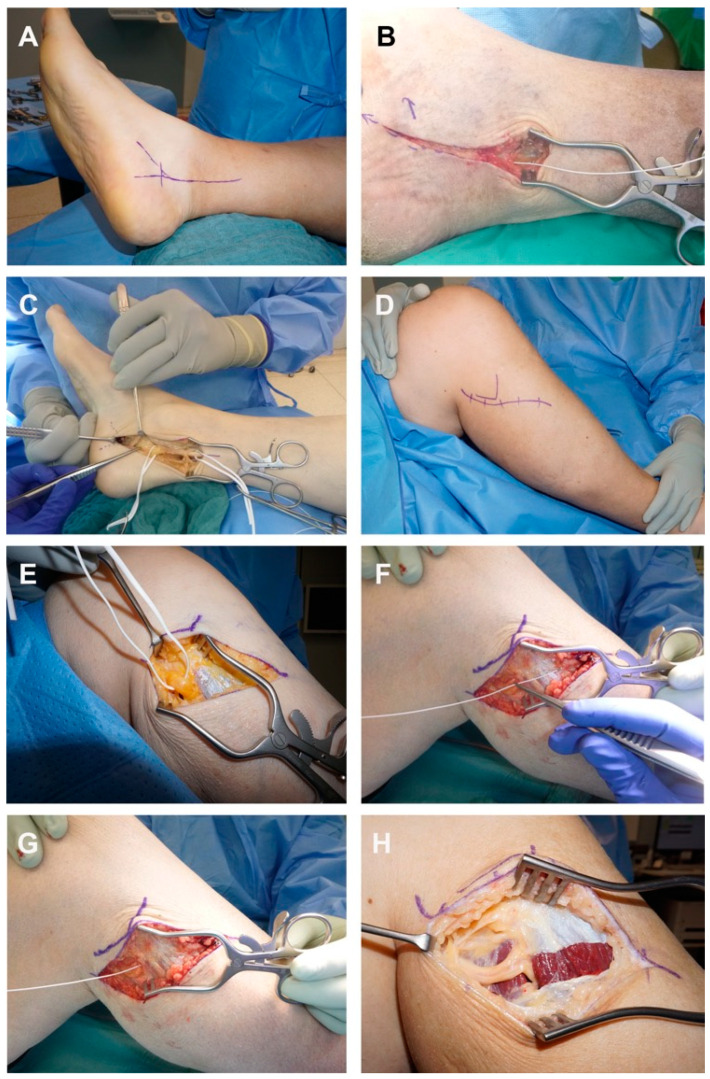
Surgical decompression of the tibial nerve (**A**–**C**) and the peroneal nerve (**D**–**H**). (**A**) Marked incision dorsal to the medial malleolus along the course of the tibial nerve. (**B**) Intraoperative compression pressure measurement in the tarsal tunnel by inserting a CE-certified flexible pressure probe (Mikro-Cath, Millar, Houston, TX, USA) between the tarsal ligament and tibial nerve. (**C**) Decompressed tibial nerve (white surgical loops) after division of the tarsal ligament and the deep fascia of the abductor hallucis muscle. (**D**) Marked curvilinear incision around the fibular neck and proximal lower leg along the course of the common peroneal nerve. (**E**) Looped common peroneal nerve with a fatty, swollen appearance proximal to the suspected fascial compression site at the posterior edge of the peroneus longus muscle. (**F**) Trajectory of the CE-certified flexible pressure probe. (**G**) Intraoperative measurement of the peroneal nerve compression pressure. (**H**) Released common peroneal nerve after division of the constricting fasciae.

**Figure 2 jpm-13-00348-f002:**
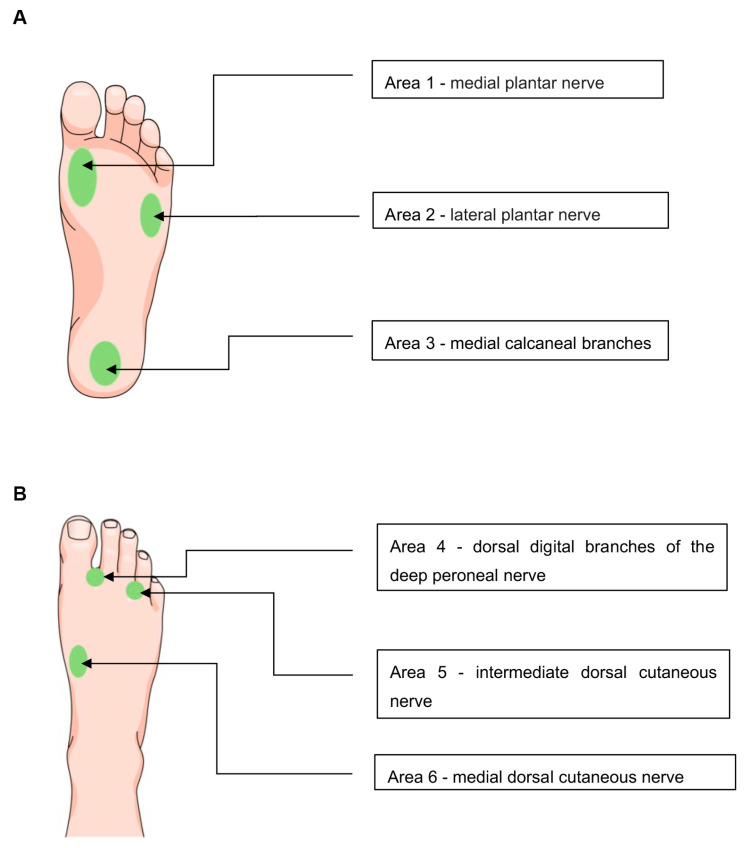
Areas of sensory testing. (**A**) In areas 1, 2, and 3, tactile thresholds and static and dynamic two-point discrimination are tested, sampling all major branches of the tibial nerve distal to the entrapment site in the tarsal tunnel. (**B**) In areas 4, 5, and 6, tactile thresholds and static and dynamic two-point discrimination are tested for the major branches of the deep and superficial peroneal nerve.

**Figure 3 jpm-13-00348-f003:**
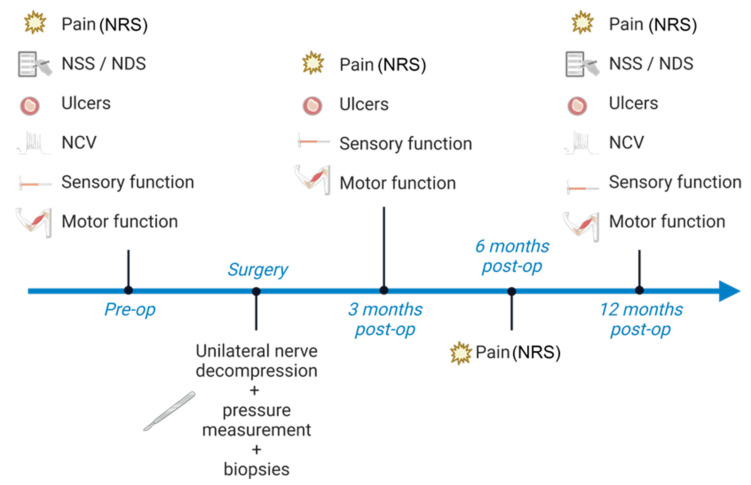
Participant timeline. Preoperatively, participants undergo detailed pain anamnesis with pain intensity scoring on the numeric rating scale (NRS), conduct the neuropathy symptom score (NSS) and neuropathy deficit score (NDS), and are screened for ulceration in the skin areas supplied by the tibial and peroneal nerve. Further, electrodiagnostic testing is conducted to measure nerve conduction velocity (NCV) and participants undergo tests for sensory and motor function. Three months post-surgery, pain intensity, ulcerations, and motor and sensory function are re-examined. Six months post-surgery, participants are asked to score their pain intensity. Twelve months post-surgery, the pre-operative set of tests is repeated.

## Data Availability

Not applicable.

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
