# Peer review of "Effect of Surgical Release of Entrapped Peripheral Nerves in Sensorimotor Diabetic Neuropathy on Pain and Sensory Dysfunction—Study Protocol of a Prospective, Controlled Clinical Trial"

_jpm, 2023, doi:10.3390/jpm13020348_

Round 1

Reviewer 1 Report

This is an interesting study that hopefully will provide evidence either for or against the use of nerve decompression for patients with diabetic peripheral neuropathy.

Regarding the patient selection criteria described in Section 3.1:

The impression that one gets reading these criteria is that the trial is trying to select for patients whose nerves are actually under pressure, i.e. who have nerve entrapment in the setting of diabetes. Unfortunately, the Authors are not giving the trial the best shot at identifying these patients. For example, the Neuropathy Deficit Score, Tinel sign, conduction block, and MRN T2 hyperintensity would presumably not distinguish a diseased nerve from a compressed nerve. Not all diabetic nerves are under pressure; some may just be metabolically “sick” and not under pressure. In practice we routinely see inflamed nerves that are electrically dysfunctional and bright on MRN T2, but that are not compressed. Sick, but not compressed, nerves would likely show abnormalities with all of these tests. 

One could argue that high resolution diagnostic ultrasound would be a much better modality for establishing whether a nerve is actually compressed or not. If a nerve is inflamed, dysfunctional, and sensitive, it will likely be described as “compressed” using the criteria in this trial. But if the nerve is all of these things, but not actually compressed, an ultrasound should easily make that determination. It would be a shame to carry out an entire clinical trial and not have a way of determining if a patient’s nerves are actually under pressure (or not) before their enrollment/surgery.

Also, I am not sure that pressure measurements alongside the nerves of an anesthetized patient wearing a tourniquet will be useable. Is there any evidence to suggest that intraoperative pressure measurements obtained in this fashion are even remotely reflective of reality for an awake, ambulatory patient?

Author Response

This is an interesting study that hopefully will provide evidence either for or against the use of nerve decompression for patients with diabetic peripheral neuropathy.

Regarding the patient selection criteria described in Section 3.1:

The impression that one gets reading these criteria is that the trial is trying to select for patients whose nerves are actually under pressure, i.e. who have nerve entrapment in the setting of diabetes. Unfortunately, the Authors are not giving the trial the best shot at identifying these patients. For example, the Neuropathy Deficit Score, Tinel sign, conduction block, and MRN T2 hyperintensity would presumably not distinguish a diseased nerve from a compressed nerve. Not all diabetic nerves are under pressure; some may just be metabolically “sick” and not under pressure. In practice we routinely see inflamed nerves that are electrically dysfunctional and bright on MRN T2, but that are not compressed. Sick, but not compressed, nerves would likely show abnormalities with all of these tests. 

One could argue that high resolution diagnostic ultrasound would be a much better modality for establishing whether a nerve is actually compressed or not. If a nerve is inflamed, dysfunctional, and sensitive, it will likely be described as “compressed” using the criteria in this trial. But if the nerve is all of these things, but not actually compressed, an ultrasound should easily make that determination. It would be a shame to carry out an entire clinical trial and not have a way of determining if a patient’s nerves are actually under pressure (or not) before their enrollment/surgery.

Response: Thank you. Indeed, identifying patients that potentially benefit from surgical release can be challenging in this cohort as electrodiagnostics may be unreliable in neuropathic nerves. In our experience predominantly focal (pain) symptoms together with the Tinel sign can be helpful, but we also heavily rely on modern magnetic resonance imaging or “MR neurography” for peripheral nerve pathologies in collaboration with the Department of Neuroradiology at Heidelberg University. This department is heavily involved in research investigating MR signal alterations in diabetic neuropathy. In the preoperative MRN we aim at identifying characteristic signal intensity alterations within nerve entrapment sites that complement our decision making whether a nerve is classified as compressed or not. In our preliminary studies, we found ultrasound to be less reliable due to interrater variability. However, in the right hands, we absolutely agree that ultrasound may be helpful for identifying nerve entrapment.

Also, I am not sure that pressure measurements alongside the nerves of an anesthetized patient wearing a tourniquet will be useable. Is there any evidence to suggest that intraoperative pressure measurements obtained in this fashion are even remotely reflective of reality for an awake, ambulatory patient?

Response: Great question. For those measurements we use a pressure sensing probe which is advanced along the nerve into the entrapment site. We do not use tourniquets for this type of surgery. We agree that the pressure exerted on the nerve in an ambulatory patient will be variable depending on muscle contraction, extremity position and others. However, we aim to quantify the compression pressure at rest in a standardized fashion for correlation with symptom severity, MRI signal characteristics and postsurgical relieve. To our knowledge, we are the first to investigate compression pressures in those patients with this device. Thus, these results need to be interpreted with great care.

Reviewer 2 Report

This is an excellent proposed study . They are planning to study very important questions  
The results will be very valuable  the study

is well designed.  However this is only 

a proposed study . As far as I can tell 

the patients have it yet been recruited .

So there are no results available to analyze. 
i expect that this study is statistically powered to provide important new information Bout the role of surgical intervention for patients with peripheral 

nerve disorders in their lower extremities 

involving the peroneal and tibial nerves  

I look forward to reading this study when it is completed .

Author Response

This is an excellent proposed study. They are planning to study very important questions  
The results will be very valuable. The study is well designed.  However, this is only a proposed study as far as I can tell the patients have it yet been recruited. So, there are no results available to analyze. iI expect that this study is statistically powered to provide important new information about the role of surgical intervention for patients with peripheral nerve disorders in their lower extremities involving the peroneal and tibial nerves  

I look forward to reading this study when it is completed.

Response: Thank you. We will report the results of this study in a future publication.

Round 2

Reviewer 1 Report

Thank you for addressing my concerns. This should be a nice study once completed.